# Evaluation of Selected Plant Volatiles as Attractants for the Stick Tea Thrip *Dendrothrips minowai* in the Laboratory and Tea Plantation

**DOI:** 10.3390/insects13060509

**Published:** 2022-05-28

**Authors:** Chunli Xiu, Fengge Zhang, Hongsheng Pan, Lei Bian, Zongxiu Luo, Zhaoqun Li, Nanxia Fu, Xiaoming Cai, Zongmao Chen

**Affiliations:** 1Tea Research Institute, Chinese Academy of Agricultural Sciences, Hangzhou 310008, China; xiuchunli@tricaas.com (C.X.); bianlei@tricaas.com (L.B.); luozongxiu@tricaas.com (Z.L.); zqli@tricaas.com (Z.L.); funanxia@tricaas.com (N.F.); 2Key Laboratory of Biology, Genetics and Breeding of Special Economic Animals and Plants, Ministry of Agriculture and Rural Affairs, Hangzhou 310008, China; 3Henan Institute of Science and Technology, Xinxiang 453003, China; zhangfengge2022@163.com; 4Institute of Plant Protection, Xinjiang Academy of Agricultural Sciences, Urumqi 830091, China; panhongsheng0715@163.com

**Keywords:** *Dendrothrips minowai*, plant-derived semiochemicals, electroantennogram activity, behavioral response, field trapping

## Abstract

**Simple Summary:**

The stick tea thrip *Dendrothrips minowai* is a key pest in tea plantations in China. In recent years, plant-derived semiochemicals have attracted considerable attention as promising attractants for the management of thrips, due to their safety and low cost. In this study, compounds that have been reported to attract other thrips or emitted from tea plants were evaluated for their electroantennogram (EAG), behavioral tests and field trapping efficacy for *D. minowai*. The EAG relative response value of *D. minowai* evoked by p-anisaldehyde, 3-methyl butanal, (E)-β-ocimene, farnesene, nonanal, eugenol, (+)-α-pinene, limonene, (−)-α-pinene, and γ-terpinene was significantly higher than the other compounds. Meanwhile, p-anisaldehyde, eugenol, farnesene, methyl benzoate, 3-methyl butanal, (E)-β-ocimene, (−)-α-pinene, (+)-α-pinene, and γ-terpinene led to attraction or repellency responses of female *D. minowai*. In addition, trap capture numbers of female *D. minowai* on sticky traps baited with p-anisaldehyde, eugenol, farnesene, and 3-methyl butanal were significantly higher than the control in tea plantations. Overall, our results highlight the potential application of plant volatiles in the development of effective, eco-friendly lure formulations for use in the monitoring and management of thrips.

**Abstract:**

The stick tea thrip (*Dendrothrips minowai* Priesner) is the main pest thrip in tea (*Camellia sinensis*) plantations in China, and seriously affects the quality and yield of tea. Plant-derived semiochemicals provide an alternative to pheromones as lures and these compounds possess powerful attractiveness. In this study, we selected 20 non-pheromone semiochemicals, including compounds that have been reported to attract other thrips and some volatiles emitted from tea plants as the potential attractant components for *D. minowai*. In electroantennogram (EAG) assays, 10 synthetic compounds (p-anisaldehyde, 3-methyl butanal, (E)-β-ocimene, farnesene, nonanal, eugenol, (+)-α-pinene, limonene, (−)-α-pinene, and γ-terpinene) elicited significant antennal responses in female *D. minowai*. In addition, a two-choice H-tube olfactometer bioassay showed that *D. minowai* displayed significant positive responses to eight compound dilutions (p-anisaldehyde, eugenol, farnesene, methyl benzoate, 3-methyl butanal, (E)-β-ocimene, (−)-α-pinene, and (+)-α-pinene) when compared with the solvent control at both 1 and 2 h. Moreover, γ-terpinene exhibited a significantly deterrent effect on *D. minowai*. Finally, trap catches of four compounds (p-anisaldehyde, eugenol, farnesene, and 3-methyl butanal, respectively) significantly increase in tea plantations. Among these, the maximum number of *D. minowai* collected by blue sticky traps baited with p-anisaldehyde was 7.7 times higher than the control. In conclusion, p-anisaldehyde, eugenol, farnesene, and 3-methyl butanal could significantly attract *D. minowai* in the laboratory and under field conditions, suggesting considerable potential as commercial attractants to control *D. minowai* populations.

## 1. Introduction

Semiochemicals are single or mixed substances emitted by one organism that stimulate a physiological or behavioral response between the same members or another species [1]. The recognition of host and non-host plants involves a complex set of physiological activities, in part mediated by semiochemicals [2]. There is no doubt that numerous semiochemical compounds released by plants could elicit significant behavioral responses of thrips [3,4]. Furthermore, semiochemicals produced by plants as well as their mimics, always provide an alternative to pheromones as lures, and sometimes can be more widely used and powerful attractants of thrips [5,6,7]. Semiochemicals produced by plants fulfill the following advantages necessary for promising lures of thrips: (a) Low toxicity to the environment, humans, and non-target beneficial arthropods, such as natural enemies; (b) volatile under normal environmental conditions; (c) stable for a long period of time; (d) the cost is relatively low compared with the established control measures; and (e) can attract more than one species of thrips [8].

Plant-derived semiochemical lures of thrips can be mainly grouped into three categories, including benzene and their derivatives, pyridine and their derivatives or other floral and fruit volatiles [4,9,10]. In particular, benzenoid compounds, such as p-anisaldehyde, methyl anthranilate, and methyl benzoate had significant attractiveness to many species of thrips [11,12,13,14]. In addition, thrips were significantly attracted by volatiles of alcohols, esters, and terpenes, such as geraniol, methyl salicylate, and β-myrcene [15,16,17]. Moreover, some natural products comprising the abovementioned plant volatiles or their derivatives have been commercialized. For instance, Lurem-TR (https://www.koppert.com/lurem-tr/) (accessed on 18 May 2022) with methyl isonicotinate (MI) as the effective ingredient has been used to control thrips [18,19]. Lurem-TR lures with colored sticky traps have been successfully used for monitoring and mass-trapping of thrips in greenhouse-grown crops, such as roses, capsicums, and sweet pepper [10].

The stick tea thrip *Dendrothrips minowai* Priesner (Thysanoptera: Thripidae) is a regional fulminant, but an easily overlooked pest of tea plants, *Camellia sinensis* (L.) O. Ktze. in China [20]. In particular, *D. minowai* have been expanded to most of the entire tea production areas and the damage degree has been intensified each year [21,22]. Both adults and nymphs of *D. minowai* feed on the leaflet of tea plants, causing stunted growth, color fading, and even stiffness or brittleness of the damaged leaves, which seriously affect the yield and quality of tea [23]. An array of tea plant volatiles is used by herbivores for host searching and locating, and several of these compounds have been investigated as non-pheromonal attractants to improve the capture of pests [24,25,26]. For instance, (Z)-3-hexen-1-ol, (Z)-3-hexenyl acetate, and linalool are volatiles derived from tea plants and significantly attract *Empoasca onukii* adults [27]. However, the compounds of tea plants that can attract *D. minowai* are not well understood.

At present, controlling the *D. minowai* population is still mainly dependent on the application of pesticides, which caused pressure on the environment and risks to beneficial natural enemies [28,29]. As a result, there is an urgent need to develop a sustainable and environmentally friendly management program that provides alternatives to synthetic insecticides. Here, we specifically addressed plant volatiles, including those that have been reported to attract other thrips and some volatiles emitted from tea plants, which are effective compounds in controlling *D. minowai* using EAG and behavioral assays in the laboratory. Then, the efficiency of these compounds were evaluated in tea gardens, which would provide new approaches to the control of thrips.

## 2. Materials and Methods

### 2.1. Biological Materials

*Dendrothrips minowai* adults were collected from tea orchards of Shaoxing royal tea village Co., Ltd., Shaoxing, China (120.71° E, 29.94° N). Thereafter, *D. minowai* were reared on tea seedling leaves (cultivar: Longjing 43) in clear glass jars in a climate chamber at 24 ± 1 °C, 60 ± 5% relative humidity, and L16⁚D8 photoperiod. In addition, fresh leaves were supplied daily to the thrips. To obtain groups of second adults of known age for the bioassays, *D. minowai* females were allowed to lay eggs on tea seedling leaves in separate glass jars for 24 h. Second generation females were used for electrophysiological recordings and behavioral bioassays.

### 2.2. Chemicals

Twenty types of compounds, derived from tea plants and plant-derived semiochemicals attracted to other thrips, were used in our trials [4,24,25,26]. In particular, 4-acetylpyridine, p-anisaldehyde, decanal, eugenol, farnesene (mixture of isomers, α-farnesene, and (E)-β-farnesene), geraniol, (Z)-3-hexenol, (Z)-3-hexenyl butyrate, limonene, methyl anthranilate, methyl benzoate, 3-methyl butanal, methyl isonicotinate, methyl salicylate, β-myrcene, nonanal, (E)-β-ocimene, (−)-α-pinene, (+)-α-pinene, and γ-terpinene were all purchased from Sigma-Aldrich (St. Louis, MO, United States) (Appendix A). Hexane (HPLC grade, CNW Technologies GmbH (Düsseldorf, Germany)) was chosen as solvent, and the abovementioned synthetic volatiles were diluted to a specific concentration.

### 2.3. Electrophysiological Recordings

An electroantennogram detection device, comprising an IDAC-2 interface box, CS-55 air stimulus controller (Syntech, Hilversum, The Netherlands) with 220 kV of electricity, was linked to a computer with EAG-Pro software. The head of a 1–3 day-old unmated female *D. minowai* individual was excised from the body using a scalpel, and connected to reference electrodes using a glass capillary. Then, the two antennae tips were immersed in a recording electrode containing electrode solutions (128 mM of NaCl, 1.9 mM of CaCl_2_, 7.6 mM of KCl, and 2.4 mM of NaHCO_3_) (see Figure 1) [30,31,32]. Mineral oil was used as the solvent to prepare the solution (10 mg/mL) of each compound, as well as the negative control. In addition, cis-3-hexen-1-ol (100 mg/mL) was utilized as the reference response, which elicited a stable antennal response of *D. minowai* (based on our preliminary experiments). Thereafter, 10 μL of the odor solution was applied on a filter paper strip (5 × 50 mm) and placed inside a glass Pasteur pipette cartridge (14.5 cm long) [33]. A constant flow of clean air (charcoal and humid filtered) was provided to the antenna to remain active at a rate of 300 mL/min. The antenna was tested at 0.5 s pulses once per stimulus. Each recording trial was conducted in the following sequence: Mineral oil, reference compound, tested odor solution (10 mg/mL), reference compound, and mineral oil, applied (0.5-s pulse) at 30-s intervals. Each chemical dilution was tested on six individuals.

### 2.4. H-Tube Olfactometer Bioassays

A glass H-tube olfactometer (See Appendix A for specific parameters) was used for testing the responses of the stick tea thrips to different plant volatiles. Prior to testing, the ends of the transverse tube, which were connected to two straight tubes, were sealed with gauze (200 mesh) [34]. Meanwhile, H-tube olfactometers were cleaned with ethanol and dried at 90 °C in an electric heating constant temperature drying oven (Shanghai Sumsung Laboratory Instrument Co., LTD., Shanghai, China) for 1 h. Tests were performed in a dark room, and both the temperature and relative humidity were maintained at 26 ± 1 °C and 65 ± 1%, respectively. In the tests, 20 μL mineral oil solution applied on a filter paper (diameter: 20 mm) was used as control, and 20 μL test odor dilution (10 mg/mL) applied on a filter paper (diameter: 20 mm) served as odor source. In general, *D. minowai* females were isolated from tea leaves into a glass jar for 2 h for acclimation and to calm down [35]. Thereafter, twenty 1–3 day-old unmated females were inoculated in the middle of the transverse tube. If the tested individuals crossed more than 5 cm of the transverse tubes, it was recorded as a positive response to the treatment or the control. If not, it was recorded as no reaction. After 1 and 2 h, we recorded the number of thrips that crossed more than and less than 5 cm of the transverse tubes, respectively. Each thrip was tested only once, and the test was repeated 10 times for each chemical. Moreover, in this study, a total of 200 *D. minowai* females were used. All of the bioassays were conducted between 08:00–17:00 h daytime.

### 2.5. Field Trapping Trials

To determine the attractiveness of active compounds produced in the laboratory to *D. minowai*, with both higher electrophysiological response value and positive behavioral response, field trapping trials were conducted in two tea gardens at Hangzhou Fuhaitang Tea Ecological Technology Co., Ltd. (120.03° E, 30.13° N) and Shaoxing Royal Tea Village Co., Ltd. (Shaoxing, China) in September 2020 and 2021. The cultivar in both tea plantations was clone Longjing 43. A red rubber septa (Zhangzhou Ingeer Agricultural Science and Technology Co. Ltd., Zhangzhou, Fujian Province, China) (117.74° E, 24.52° N) comprising 100 μL of compound dilution (concentration: 100 mg/mL, 10 mg active compound) or solvent (hexane) solution was placed on a blue sticky trap. Each compound dilution was replicated 6 times, for a total of 48 blue sticky traps. These traps were placed with 6 m spacing between traps to minimize the interference between lures. Prior to the trapping trials, blue sticky traps without any lures were placed at the trap sites for 2 days to confirm that the thrip populations were at approximately equal population densities in the different treatment areas (Hangzhou: 30 August 2020; Shaoxing: 31 August 2021) [36]. The trap check was conducted after 2 days of experiment, and trap catches were recorded. Meanwhile, the traps were replaced on each sampling day, but the lures were not replaced during the experiment. Each volatile treatment was replicated 3 times in 2020 and 2021.

### 2.6. Statistical Analyses

All of the data were checked for normality and equality of variances prior to statistical analysis. Datasets that did not fit assumptions were square-root (sqrt) transformed to meet the requirements of equal variance and normality. The EAG relative value was calculated based on the following: EAG amplitude values of tested stimulus—the average EAG amplitude values of the control)/(the average EAG amplitude values of the reference compound—the average EAG amplitude values of the control) × 100%. Comparisons of recorded EAG values between different compounds were performed by one-way ANOVA following Tukey’s HSD test (*p* < 0.05). In the behavioral trial, the *t*-test was applied to assess the number of thrip individuals between the treatment and control (*p* < 0.05). In the field evaluation, we performed a general linear model (GLM) analysis to assess the effect of various active compounds on thrips. Dunnett’s one-tailed *t*-test was controlled for Type 1 experiment-wise error for comparisons of each compound against control (a Duncan test with α = 0.05). The graph of EAG relative value was performed using the GraphPad Prism 7.0. The graph of behavioral response and the field evaluation were performed using OriginPro 2021. All of the analyses were conducted using SAS 9.4.

## 3. Results

### 3.1. EAG Response of D. minowai Adults to Single Compound

To comprehensively evaluate the electrophysiological activity, EAG responses of *D. minowai* to 20 compounds were tested. For a specific concentration (10 mg/mL), 10 volatiles elicited significant antennal responses to female *D. minowai*, with the mean relative values (±SEM) approximately equal to 1.00 (p-anisaldehyde: 1.48 ± 0.046; 3-methyl butanal: 1.43 ± 0.040; (E)-β-ocimene: 1.31 ± 0.121; farnesene: 1.25 ± 0.093; nonanal: 1.17 ± 0.059; eugenol: 1.15 ± 0.040; (+)-α-pinene: 1.11 ± 0.057; limonene: 1.08 ± 0.072; (−)-α-pinene: 0.96 ± 0.061; γ-terpinene: 0.95 ± 0.036). Whereas 10 other compounds elicited weak antennal responses. Among these, p-anisaldehyde elicited the strongest EAG response to *D. minowai* (*p* < 0.001) (Figure 2).

### 3.2. Behavioral Response to Single Compound

In a dual-choice assay, *D. minowai* showed a significant preference for 8 compound dilutions in comparison with the solvent control at both 1 and 2 h (p-anisaldehyde: 1 h: *F*_1,18_ = 562.37, *p* < 0.001; 2 h: *F*_1,18_ = 477.18, *p* < 0.001; eugenol: 1 h: *F*_1,18_ = 204.545, *p* < 0.001; 2 h: *F*_1,18_ = 202.96, *p* < 0.001; farnesene: 1 h: *F*_1,18_ = 244.45, *p* < 0.001; 2 h: *F*_1,18_ = 115.04, *p* < 0.001; methyl benzoate: 1 h: *F*_1,18_ = 125.29, *p* < 0.001; 2 h: *F*_1,18_ = 175.02, *p* < 0.001; 3-methyl butanal: 1 h: *F*_1,18_ = 396.37, *p* < 0.001 (Figure 3a); 2 h: *F*_1,18_ = 368.01, *p* < 0.001; (E)-β-ocimene: 1 h: *F*_1,18_ = 113.14, *p* < 0.001; 2 h: *F*_1,18_ = 82.79, *p* < 0.001; (−)-α-pinene: 1 h: *F*_1,18_ = 132.81, *p* < 0.001; 2 h: *F*_1,18_ = 82.90, *p* < 0.001; (+)-α-pinene: 1 h: *F*_1,18_ = 132.25, *p* < 0.001; 2 h: *F*_1,18_ = 104.73, *p* < 0.001) (Figure 3b). In contrast, the thrips were not significantly attracted by 11 compounds at both 1 and 2 h (4-acetylpyridine, decanal, geraniol, (Z)-3-hexenol, (Z)-3-hexenyl butyrate, limonene, methyl anthranilate, methyl isonicotinate, methyl salicylate, β-myrcene, and nonanal, all *p* > 0.05). Of note, γ-terpinene exhibited a significantly deterrent effect to *D. minowai* (1 h: *F*_1,18_ = 42.29, *p* < 0.001; 2 h: *F*_1,18_ = 41.51, *p* < 0.001).

### 3.3. Field Evaluation of Seven Compounds to Adult Thrips

Blue sticky traps baited with seven electrophysiological and behavioral active plant volatiles (p-anisaldehyde, eugenol, farnesene, 3-methyl butanal, (E)-β-ocimene, (−)-α-pinene, and (+)-α-pinene) were tested at Hangzhou (in 2020) and Shaoxing (in 2021). Prior to the real trapping trials, *D. minowai* populations were at approximately equal population densities for different treatments (Hangzhou: *F*_1,10_ = 0.22, *p* = 0.977; Shaoxing: *F*_1,10_ = 0.54, *p* = 0.807) (Figure 4). At Hangzhou, the traps baited with four individual compounds (p-anisaldehyde, eugenol, farnesene, and 3-methyl butanal) captured significantly more *D. minowai* adults than control, respectively (p-anisaldehyde: 2020/9/1: *F*_1,10_ = 1259.321, *p* < 0.0001, 2020/9/3: *F*_1,10_ = 316.825, *p* < 0.0001, 2020/9/5: *F*_1,10_ = 340.533, *p* < 0.0001; eugenol: 2020/9/1: *F*_1,10_ = 449.733, *p* < 0.0001, 2020/9/3: *F*_1,10_ = 235.42, *p* < 0.0001, 2020/9/5: *F*_1,10_ = 216.161, *p* < 0.0001; farnesene: 2020/9/1: *F*_1,10_ = 100.873, *p* < 0.0001, 2020/9/3: *F*_1,10_ = 45.77, *p* < 0.0001, 2020/9/5: *F*_1,10_ = 155.773, *p* < 0.0001; 3-methyl butanal: 2020/9/1: *F*_1,10_ = 137.612, *p* < 0.0001, 2020/9/3: *F*_1,10_ = 354.862, *p* < 0.0001, 2020/9/5: *F*_1,10_ = 734.221, *p* < 0.0001). Among these, the number of thrips collected by blue sticky traps baited with p-anisaldehyde were the highest by 5.0, 5.0, and 7.7 times of control, respectively (Figure 4a; Appendix A). Similarly, traps baited with four individual compounds captured significantly more *D. minowai* adults than control at Shaoxing, respectively. Moreover, the number of thrips collected by traps baited with p-anisaldehyde were the highest at Shaoxing by 5.4, 4.8, and 3.7 times of control, respectively (Figure 4b; Appendix A).

## 4. Discussion

Plants constantly emit complex blends of volatile odors, which provide cues for herbivores [37]. The application of plant volatiles to attract and kill pests is an important part of integrated pest management (IPM) strategies [38]. In the future, the electrophysiological and behavioral response experiments in the laboratory are an important basis to test and utilize these volatile odors for trapping in field [4]. We screened seven electrophysiological and behavioral active volatiles of *D. minowai* using EAG detection and H-tube olfactometer bioassays. Furthermore, four compounds significantly increase the trap captures of *D. minowai* in tea garden, and the trapping effect was relatively better than those of other thrips. This work enriched the research on plant-derived semiochemicals for use in attracting and killing non-flowering thrips, which will promote the development of the commercial attractant products.

Generally, the behavioral response of flower-inhabiting thrips has been studied extensively, indicating that flower-inhabiting thrips respond to pyridine and benzene [4,10], whereas the non-flowering thrips do not respond to these dilutions [39]. For example, methyl isonicotinate (MI) and p-anisaldehyde elicit stronger behavioral responses to more than 10 flower-inhabiting species of thrips [4,14,40]. Similar to previous reports, we found that non-flower-inhabiting *D. minowai* do not respond to MI and 4-acetylpyridine dilutions in this study. However, contrary to previous reports, it was demonstrated that D. minowai exhibited a highly significant preference to p-anisaldehyde and methyl benzoate dilutions in the present study. In addition, γ-terpinene emitted from tea plants had significant avoidance effects on *D. minowai* in the laboratory, which showed that the repellent properties of γ-terpinene to *D. minowai* would be helpful in the development of push–pull strategy.

At present, there are few reports regarding the electrophysiological trials of thrips. It is hypothesized that the difficulty of electrophysiological trials may be due to the small antenna of thrips. Coupled gas chromatography–electroantennogram detection (GC–EAD) showed that one component of male *Megalurothrips usitatus* specific odor namely (2E, 6E)-farnesyl acetate elicited a significant electrophysiological response to *M. usitatus* [41]. In this study, we screened 11 synthetic compounds, including p-anisaldehyde, eugenol, farnesene, limonene, methyl isonicotinate, 3-methyl butanal, nonanal, (E)-β-ocimene, (−)-α-pinene, (+)-α-pinene, and γ-terpinene, that elicited a significant antennal response to female *D. minowai*. These results determine that *D. minowai* rely on more than one component to search for host plants, and multiple components work together in locating host plants. This phenomenon has been studied extensively in other herbivores. For example, three *Adelphocoris* species responded to seven compounds in EAG trials from 11 host plants [31].

Research on plant-derived semiochemicals for thrips could promote the development of efficient and novel measures as alternatives to synthetic insecticides [4,29]. Generally, non-pheromone semiochemicals can trap both sexes. However, in our trials, few male *D. minowai* were found to be attracted to traps. This could be due to the fact that male *D. minowai* fly fast, their behaviors are likely to be more complex, and they rely on many plant cues (color, odor, shape, etc.) [42]. In our trials, the maximum number of female *D. minowai* collected by blue sticky traps baited with p-anisaldehyde was 7.7 times higher than hexane. The trapping effect of p-anisaldehyde on *D. minowai* was relatively better than the other thrips, which suggested that p-anisaldehyde has a great application prospect. For example, significant 2.4–3.9 × increases in *Frankliniella occidentalis* numbers in traps with MI when compared with those without MI were found during the whole season in a nectarine orchard [6], and the number of *F. occidentalis* on the sticky cards baited with the commercial non-pheromone semiochemical product Lurem-TR could be increased by 4 times [15]. Occasionally, a specific proportion of compound mixture is more attractive than single volatiles [43,44,45]. For instance, traps with both ethyl nicotinate and ethyl isonicotinate lures caught 2–3 times higher numbers of adult New Zealand flower thrips than traps with ethyl isonicotinate only [46]. Binyameen et al. [47] observed a significant additive trapping effect for the response of onion thrips (*Thrips tabaci*) to eugenol and ethyl isonicotinate combined. Therefore, a certain fieldwork and laboratory evaluation is required on an urgent basis in the near future, which would provide theoretical reference for the development of *D. minowai* attractants.

## 5. Conclusions

In conclusion, we screened seven electrophysiological and behavioral active non-pheromone semiochemicals of *D. minowai* using EAG detection and H-tube olfactometer bioassays. In the fields, we demonstrated that the positive effect of p-anisaldehyde, eugenol, farnesene, and 3-methyl butanal as attractants of *D. minowai* caused increased efficiency in catching traps. In particular, the maximum number of *D. minowai* trapped by blue sticky traps baited with p-anisaldehyde was 7.7 times higher than the control. In the future, this work will lead to the development of plant-derived semiochemicals for use in monitoring and mass trapping of *D. minowai*, which may have potential applications in the integrated management of tea thrips.

## Figures and Tables

**Figure 1 insects-13-00509-f001:**
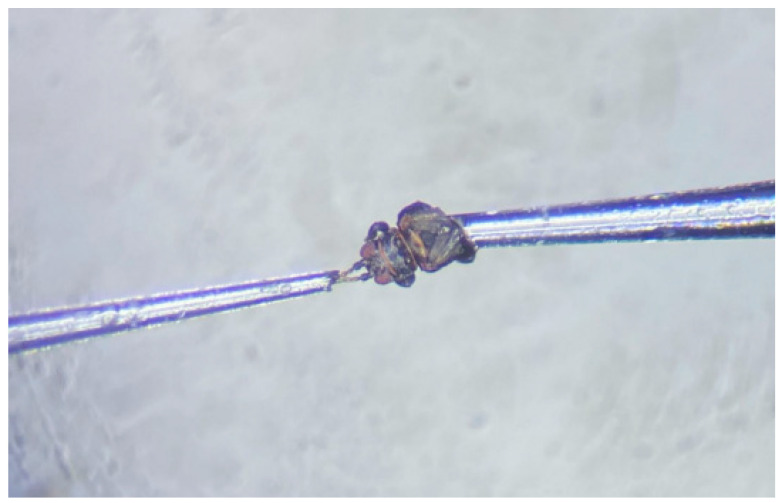
Antenna connection mode of *Dendrothrips minowai* in electroantennogram (EAG) trials.

**Figure 2 insects-13-00509-f002:**
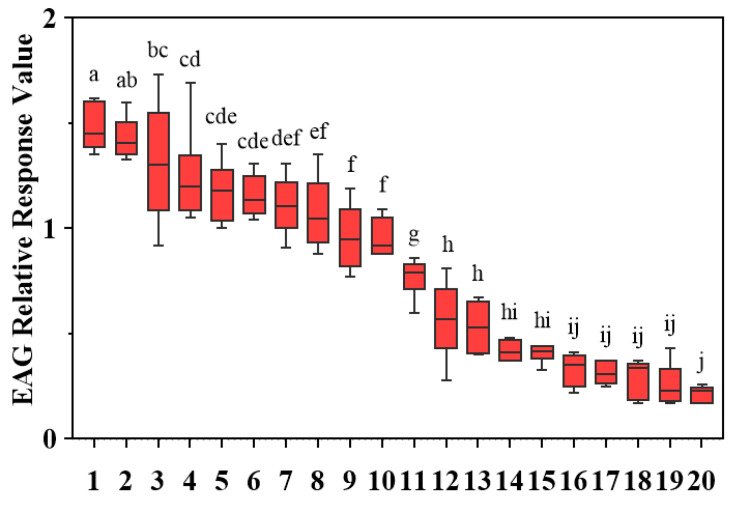
EAG relative response values of female *Dendrothrips minowai* to synthetic compounds. Means (±SEM) indicated by the same letter are not significantly different (*p* > 0.05), and those indicated by different letters are significantly different (*p* < 0.05). 1: P-anisaldehyde; 2: 3-Methyl butanal; 3: (E)-β-ocimene; 4: Farnesene; 5: Nonanal; 6: Eugenol; 7: (+)-α-Pinene; 8: Limonene; 9: (−)-α-pinene; 10: γ-Terpinene; 11: β-Myrcene; 12: (Z)-3-hexenol; 13: (Z)-3-hexenyl butyrate; 14: Methyl salicylate; 15: Methyl benzoate; 16: Methyl isonicotinate; 17: Methyl anthranilate; 18: Geraniol; 19: Decanal; 20: 4-Acetylpyridine.

**Figure 3 insects-13-00509-f003:**
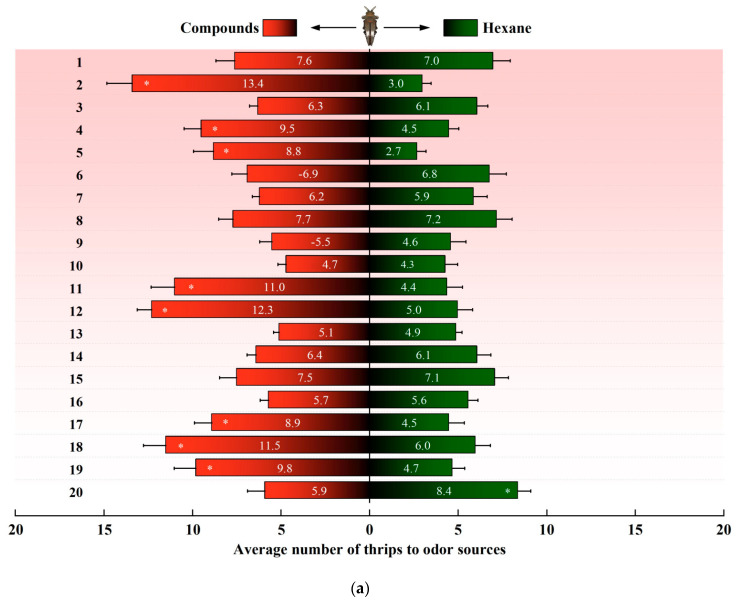
Preferences of female *Dendrothrips minowai*. (**a**) Observation for 1 h; (**b**) observation for 2 h. Choice of *D. minowai* when offered control and different volatile compounds in an H-tube olfactometer. Bars represent the average number of thrips choosing either of odor sources. 1: 4-Acetylpyridine; 2: P-anisaldehyde; 3: Decanal; 4: Eugenol; 5: Farnesene; 6: Geraniol; 7: (Z)-3-hexenol; 8: (Z)-3-hexenyl butyrate; 9: Limonene; 10: Methyl anthranilate; 11: Methyl benzoate; 12: 3-Methyl butanal; 13: Methyl isonicotinate; 14: Methyl salicylate; 15: β-Myrcene; 16: Nonanal; 17: (E)-β-ocimene; 18: (−)-α-pinene; 19: (+)-α-Pinene; 20: γ-Terpinene. “*” denotes a significant difference at *p* < 0.05 level. For each compound, 200 individuals of female thrips were tested.

**Figure 4 insects-13-00509-f004:**
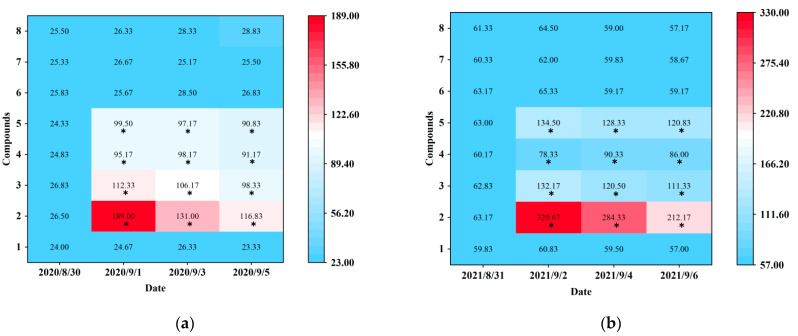
Captures of female *Dendrothrips minowai* on blue sticky traps baited with control and different compounds. (**a**) Hangzhou in 2020; (**b**) Shaoxing in 2021. Mean captures are shown from six replicates. 1: Hexane (solvent control); 2: P-anisaldehyde; 3: Eugenol; 4: Farnesene; 5: 3-Methyl butanal; 6: (E)-β-ocimene; 7: (−)-α-pinene; 8: (+)-α-Pinene. “*” denotes a significant difference at *p* < 0.05 level.

## Data Availability

All data analyzed in this study are included in this article.

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
