# Peer review of "Evaluation of Selected Plant Volatiles as Attractants for the Stick Tea Thrip *Dendrothrips minowai* in the Laboratory and Tea Plantation"

_insects, 2022, doi:10.3390/insects13060509_

Round 1

Reviewer 1 Report

The main research question on
Evaluation of Selected Plant Volatiles as Attractants for Dendrothrips minowai in the Laboratory and Tea Plantation
And is exactly on the tea thrips pest (Dendrothrips minowai Priesner)

2. Do you consider the subject to be original or related to the field, and if so, why?

Yes, according to the study and review of the submitted article, the subject is related to the field of entomology and there are few articles with this title. And I find this article suitable for publication because of the careful analysis of 11 compounds.

3. What adds to the subject area compared to other published material?
This article introduces a new formulation for controlling thrips, and is very important compared to the chemical toxins used in the market.

3. What adds to the subject area compared to other published material?
This article introduces a new formulation for controlling thrips, and is very important compared to the chemical toxins used in the market.
What voltage is used in section 2-3? From 220 electricity? Or 110?

Conclusions and sources are well described.

Regards 

Reviewer 2 Report

The manuscript is interesting and has a lot of work in laboratory and in the field test, but there are many questions that must be addressed before publishing. I suggest major revisions.

Introduction

- Those listed properties (lines) about “non-pheromone semiochemicals” (allellochemicals therefore) are very generalized and must be taken with great caution, since some overlap with the advantages of the use of pheromones, or they do not have to have an effect on both males and females (e) (for example, a kairomone may only be effective on females or males). By the way, the term kairomona is not referred to at any time in the manuscript. From the reference the authors quote, it seems that they are quoting the advantages of using methyl isonicotinate.

-The authors do not cite articles related with Tubuliferous pest thrips as Liothrips jatrophae:

Behavioural and electrophysiological responses of Liothrips jatrophae (Thysanoptera: Phlaeothripidae) to conspecific extracts and some of its identified compounds

Johana González-Orellana, Guillermo López-Guillén, Edi A. Malo, Arturo Goldarazena, Leopoldo Cruz-López Phisiological Entomology First published: 06 July 2021 https://doi.org/10.1111/phen.12367

Biological Materials

- Línea 102 – “adults of known age…”, y solo figura la edad en ensayos EAG. Las hembras de ensayos comportamiento también 1-3 d old?. A su vez, no se hace alusión al estado reproductivo de las hembras usadas en los ensayos.

- Line 102 – “adults of known age…”, and only the age appears in EAG tests. Behavioral trial females are also 1-3 days old?. In turn, no reference is made to the reproductive status of the females used in the tests.

- The specimens were raised in tea leaves. Were they allowed to be isolated from these prior to plant volatile tests? May be females can be habituated to odors and mask subsequent response to volátiles.

-Males have not been tested why? Males can be completely different behaviour and are frequent in orchards in tea orchards in China.

Chemicals  

  • In the Chemicals subsection, purity and stereochemistry should at least be specified where appropriate (example farnesene, it can be (E,E), (Z,Z), etc). I see that Table S1- related information of standard compounds is mentioned, perhaps this information is there (although the use of the term standard is confusing).
  • Occimene, no Occimenne….

EAG

  • Details are missing (how many puffs of each stimulus? Why aren't negative controls sandwiched between the composite and standard stimuli??)
  • Figure 2: The compound 13 methyl isonicotinate appears in the results with a mean relative response value of 1.19, while in the graph it is one of the compounds with a lower value.
  • Another point to consider is that the response of cis-3-hexenol (compound 7) shown in Figure 2 does not seem consistent. If this is the compound that is considered as the reference (line 124), when normalizing the response according to to the suggested formula (line 173)? It would have to be 1.
  • figure 2 is quite improvable. I do not see much sense in ordering the compounds on the abscissa axis in alphabetical order. The normal thing would be to represent the data according to the magnitude of the response (higher to lower or vice versa).
  • The picture has no informative at all.

Behaviour

. Statistical treatment of olfactometry data. Why ANOVA when really comparing the response of compound vs. hexane? It doesn't make sense, better apply Chi-square

In the Figure, you would place the asterisk next to it (left in the case of attractive compound, right in the case of ϒ-terpinene). They seem silly things but I know people who still have a hard time interpreting these graphs.

Field trials

  • It would be convenient to indicate the release rate of each compound, at least in Material and Methods
  • I suspect that table 2 shows the pairwise comparisons between the different compounds used in the traps. However, it should then be pointed out that the asterisk in Figure 4 alludes to significant differences with respect to the control traps.
  • The writing could be improved. Confusing sentences, Example: line 235 “traps baited with four compounds captured significantly more…” implies that there were traps with quaternary mixtures.

Round 2

Reviewer 2 Report

The manuscript has been improved and it is a nice paper that deserves to be published in Insects